

# Herbicide injury induces DNA methylome alterations in *Arabidopsis*

Gunjune Kim[1,*], Christopher R. Clarke[1,2,*], Hailey Larose[1],
Hong T. Tran[3], David C. Haak[1], Liqing Zhang[3], Shawn Askew[1],
Jacob Barney[1] and James H. Westwood[1]

[1] Department of Plant Pathology, Physiology and Weed Science, Virginia Tech, Blacksburg, VA, USA
[2] Genetic Improvement of Fruits and Vegetables Laboratory, United States Department of Agriculture, Agricultural Research Service, Beltsville, MD, USA
[3] Department of Computer Science, Virginia Tech, Blacksburg, VA, USA
* These authors contributed equally to this work.

## ABSTRACT

The emergence of herbicide-resistant weeds is a major threat facing modern agriculture. Over 470 weedy-plant populations have developed resistance to herbicides. Traditional evolutionary mechanisms are not always sufficient to explain the rapidity with which certain weed populations adapt in response to herbicide exposure. Stress-induced epigenetic changes, such as alterations in DNA methylation, are potential additional adaptive mechanisms for herbicide resistance. We performed methylC sequencing of *Arabidopsis thaliana* leaves that developed after either mock treatment or two different sub-lethal doses of the herbicide glyphosate, the most-used herbicide in the history of agriculture. The herbicide injury resulted in 9,205 differentially methylated regions (DMRs) across the genome. In total, 5,914 of these DMRs were induced in a dose-dependent manner, wherein the methylation levels were positively correlated to the severity of the herbicide injury, suggesting that plants can modulate the magnitude of methylation changes based on the severity of the stress. Of the 3,680 genes associated with glyphosate-induced DMRs, only 7% were also implicated in methylation changes following biotic or salinity stress. These results demonstrate that plants respond to herbicide stress through changes in methylation patterns that are, in general, dose-sensitive and, at least partially, stress-specific.

## INTRODUCTION

The development of herbicide-resistant weed populations is a major crisis facing modern agriculture (*Bonny, 2016*). Herbicide resistance has evolved in populations of at least 470 different weeds, including 35 species that have developed resistance to the herbicide glyphosate (*Heap, 2016*), the most widely used herbicide in the history of agriculture (*Duke & Powles, 2008*). Mechanisms of herbicide resistance are easily explained in some cases, as when a mutation occurs in the protein target of the herbicide that reduces herbicide binding (*González-Torralva et al., 2012*) or even the copy number of herbicide

Corresponding author
James H. Westwood,
westwood@vt.edu

target genes (*Gaines et al., 2010*). Other cases, termed non-target site resistance, are poorly understood, but have been attributed to quantitative accumulation of minor resistance alleles selected from standing genetic variation at multiple gene loci (*Busi, Neve & Powles, 2013*; *Délye, 2013*). Non-target site resistance may involve various mechanisms that affect herbicide metabolism or translocation (*González-Torralva et al., 2012*). Resistance to glyphosate is interesting in that it often appears to involve non-target site mechanisms, which can emerge and dominate a population after as few as three generations of sub-lethal exposure to the herbicide (*Busi & Powles, 2009*), and can spread through a population faster than predicted by gene flow or propagule dispersal (*Escorial et al., 2011*; *Espeby, Fogelfors & Milberg, 2011*; *Okada et al., 2013*). Given that herbicides induce a strong abiotic stress, it is likely that weeds respond by activating stress-signaling networks that reprogram gene expression (*Busi, Neve & Powles, 2013*), and we hypothesize that this involves epigenetic regulation of gene function that could contribute to herbicide resistance as implicated in weed adaptation to other stresses (*Verhoeven et al., 2010*) and in the response of rice to the pesticide atrazine (*Lu et al., 2016*).

A prominent mechanism of epigenetic regulation of gene expression is through cytosine methylation. Methylated cytosines (mCs) occur in three different sequence contexts in plants: CG methylation, which is the predominant form of methylation in gene-rich regions, and CHH or CHG (H = A, T, or C) methylation predominantly in transposable elements and repetitive sequences (*Cokus et al., 2008*). CHH and CHG mCs are much more common in plants than animals (*Cokus et al., 2008*). Multiple RNA-dependent methylation pathways control *de novo* methylation in the mustard weed *Arabidopsis thaliana* (*Matzke & Mosher, 2014*) with methylation at the different sequence contexts altered and maintained through both overlapping and sequence-context-specific mechanisms. Mutations in the genes of these pathways lead to aberrations in the methylome (*Stroud et al., 2013*). Additionally, wildtype *A. thaliana* plants accumulate epimutations (i.e., changes in the methylome) over generations of greenhouse propagation (*Becker et al., 2011*; *Schmitz et al., 2011*). Stress exposure can lead to substantial methylome reprogramming as observed in *A. thaliana* following pathogen attack or salicylic acid treatment (*Dowen et al., 2012*), phosphate starvation (*Secco et al., 2015*; *Yong-Villalobos et al., 2015*), and salinity stress (*Jiang et al., 2014*; *Wibowo et al., 2016*). The majority of methylation changes following these stresses were considered transient—not transgenerationally stable—but a subset of stress-responsive methylations is stably fixed in populations over several subsequent generations of exposure to stress (*Jiang et al., 2014*). Thus, transgenerational epigenetic changes in gene expression could lead to enhanced and adaptive stress tolerance. However, the ability of epigenetic changes to be transgenerationally stable and affect the adaptation of plants in their environment remains contentious (*Hagmann et al., 2015*; *Kawakatsu et al., 2016*).

Because the specificity of methylome reprogramming following stress may appear inexact [e.g., compare *Secco et al. (2015)* to *Yong-Villalobos et al. (2015)*] and the effect of alterations in methylation levels on gene expression may seem inconsistent (*Karan et al., 2012*; *Kawakatsu et al., 2016*; *Li et al., 2012*), we tested whether there was a dosage effect by glyphosate stress on specific alterations of the plant methylome.

Glyphosate has been previously shown to alter DNA methylation in wheat (*Nardemir et al., 2015*) using relatively low resolution techniques. In addition, the herbicide atrazine was recently shown to induce global changes in methylation in rice (*Lu et al., 2016*). We used bisulfite sequencing to determine the effect of glyphosate on the *Arabidopsis* methylome at single base pair resolution to identify specific genetic loci that have altered methylation patterns following glyphosate exposure.

## MATERIALS AND METHODS

### Plant growth conditions and herbicide treatment

Seeds of *A. thaliana* ecotype Columbia were sown in Sunshine Number 1 media and stratified for three days in the dark at 4 °C. The flats were then transferred to a Conviron growth chamber with a 12 h light cycle and light intensity of 90 $\mu$mol m$^{-2}$ s$^{-1}$ and allowed to grow to a fully developed rosette, pre-floral shoot stage. Blocks of four plants were then randomly assigned treatment of glyphosate at 0%, 5%, 10%, or 15% of the label rate (0.9 kg acid equivalency (ae) ha$^{-1}$ of RoundUp Pro Concentrate). Glyphosate-treated plants were sprayed at 187 L ha$^{-1}$ in a spray booth. Following glyphosate treatment, plants were transferred to a growth shelf with a 12 h light cycle and light intensity of 90 $\mu$mol m$^{-2}$ s$^{-1}$ and grown until fully developed siliques were formed (approximately two weeks for the 0% and 5% glyphosate-treated plants and eight weeks for the 10% glyphosate-treated plants). Thus, tissues were harvested at equivalent developmental stages, despite slower growth rate of glyphosate-treated plants.

### Genomic DNA isolation and methylC sequencing library preparation

Genomic DNA was isolated from two to three cauline leaves formed following glyphosate exposure from individual plants in quadruplicate for each of the three treatment levels (0%, 5%, and 10%) using the Biosprint-15 plant DNA extraction kit (Qiagen, Hilden, Germany). The 12 samples were sent to Genomics Research Laboratory at Biocomplexity Institute of Virginia Tech (Blacksburg, VA, USA) for library preparation and bisulfite sequencing. 100 ng of intact DNA was bisulfite converted using EZ DNA Methylation-Gold Kit (#D5005; Zymo Research, Irvine, CA, USA), following the manufacturers protocol, except eluting into 9 ul. The entire amount of the purified bisulfite-treated DNA was converted to Illumina DNA libraries using EpiGnome Methyl-Seq kit (Epicentre, Madison, WI, USA). Six samples each were individually barcoded, quantitated by qPCR, and pooled to sequence on the entire Illumina HiSeq Rapid Run flowcell. Libraries were clustered on-board at a concentration of 8.5 pM with 3% phiX, onto a flow cell using Illumina's HiSeq Rapid Paired End Cluster Kit V2 (PE-402-4002), and sequenced 2× 101 cycles using HiSeq Rapid SBS Kit (200-cycles) (FC-402-4021).

### MethylC sequencing data analysis

The sequencing reads were subject to pre-processing quality control using FastQC to eliminate adapter sequences and barcodes using Trimmomatic (http://www.usadellab.org/cms/) and FastX Tookit (http://hannonlab.cshl.edu/fastx_toolkit). Low quality reads (quality score $Q < 30$) were discarded and only reads passing the quality check were

mapped to Col-0 *A. thaliana* (TAIR 10) reference genome using Bismark aligner (v 0.14.5) under default parameters (−n 1 −l 50) (*Krueger & Andrews, 2011*). Cytosine methylation information was extracted from aligned reads using Bismark methylation extractor and methylation calls for CG, CHH, and CHG contexts were generated. The conversion efficiency of bisulfite treatment (methylation status of cytosine) was estimated from reads mapped to the chloroplast genome, which is expected to be unmethylated.

## Calling DMRs

### Methylkit/eDMR

Differentially methylated regions (DMRs) between treated and control plants were identified using the R (v3.0.3) package methylkit/eDMR (*Li et al., 2013*). Methylkit allows parameter adjustment to identify DmCs based on $q$-value, percent methylation difference, and types of methylation (hyper or hypo) using statistical tests such as logistic regression and Fisher's exact test. Pairwise Pearson's correlation coefficient and Hierarchical clustering (Ward's method, correlation distance metric) were calculated based on percent methylation values for all 12 samples. Differential methylation between treated (5% and 10%) and control groups were determined using Fisher's exact test with a minimum 25% difference in methylation ratio between groups and $q$-value <0.01. DMRs were constructed using weighted optimization algorithm eDMR.

### bsseq

Differential methylation between glyphosate-treated and control libraries was also determined in R using bsseq package (bioconductor). The coverage (.cov) files generated by Bismark were used to run methylation smoothing which generates per-CG, CHH, and CHG methylation values based on at least one biological replicate for control and treated samples using mean $t$-statistics. DMRs were filtered by areaStat which was weighted by the number of methylation sites for each context (*Hansen, Langmead & Irizarry, 2012*). We identified the overlapping DMR regions for each context between percent of glyphosate exposure using the intersectBed function within the bedtools suite with default parameters (*Quinlan & Hall, 2010*). The stringency of the parameters to use when calling DMRs was optimized through analyzing the effect of increasing the number of analyzed replicates on the number of identified DMRs by calling DMRs using all possible combinations of two replicates, three replicates, and four replicates from our methylC-seq data. For the less stringent parameters (see Fig. S1 legend) relying on two replicates likely leads to many false positives because of the sharp decline in the number of DMRs when increasing the number of replicates (Fig. S1). When using the more stringent parameters, the overall number of DMR calls is lower and increasing replicates has no effect on raw number of called DMRs suggesting fewer false positives. We, therefore, used the DMRs called using the more stringent parameters for downstream analyses.

## Gene ontology enrichment analysis

Annotations from the TAIR10 database were assigned to DMRs using a custom Perl script (script provided in supplementary materials) in which genomic features were associated

with DMRs that overlapped within 2Kb in either direction. *Arabidopsis thaliana* ID lists from associated DMRs were processed and analyzed in VirtualPlant 1.3 software (http://virtualplant.bio.nyu.edu/cgi-bin/vpweb/) using the BioMaps module ($p \leq 0.05$) to find significantly over-represented gene ontology (GO) categories. To select significant functional categories, we set a cutoff point in a normalized frequency (relative frequency of input gene list/relative frequency of reference) that was greater than 1.5-fold.

## Classification of dose-dependent response of DMRs identified in both 5% and 10% datasets

Statistical hypothesis tests using the Student's *t*-test with 5% significance levels were conducted to determine if the mean difference in 5% glyphosate vs. control group ($\Delta_5$) is significantly different from the mean difference in 10% glyphosate vs. control group ($\Delta_{10}$) for each DMR. We then fitted one nonlinear curve for sites in which $\Delta_5$ is significantly greater than $\Delta_{10}$ and all DMRs below the fitted blue curve in hypermethylation case and above the fitted blue curve in hypomethylation case were classified as inverse dose-dependent (see Fig. S2). Another nonlinear curve for sites in which $\Delta_5$ is significantly less than $\Delta_{10}$ was then fitted. All DMRs above the fitted green curve in hypermethylation case and below the fitted green curve in hypomethylation case were classified as positive dose-dependent. Nonlinear curves are appropriate for fitting the data based on both Akaike information criterion (AIC) and Bayesian information criterion (BIC). The nonlinear curve fitting AIC and BIC was smaller compared to linear fitting.

## Statistical comparison of overlapping DMR-associated genes across different stresses (glyphosate, phosphate starvation, and biotic)

The number of DMR-associated genes following glyphosate treatment, phosphate starvation and biotic stress are 3,680, 712, and 884, respectively (Dataset S7). Only 13 genes were identified as being differentially methylated following all these stresses (Dataset S7). We considered the possibility that the lack of shared DMR-associated genes among all three stresses is evidence for DMRs being randomly distributed across the genome as opposed to selective in response to stress. The total number of genes in the TAIR10 annotation of the *A. thaliana* genome is 33,602. We included the total number of *A. thaliana* genes as the baseline rather than previously published lists of methylated *A. thaliana* genes because a majority of our genic DMRs (73%) were identified in genes previously characterized as unmethlyated. The glyphosate-induced DMR-associated genes account for 11% of all possible genes. If these 3,680 genes are randomly distributed across the genome, and the 884 biotic stress DMR-associated genes are likewise random, then the expected number of overlapping genes can be calculated by $0.11 \times 884 = 97$. Using the same formula for the phosphate starvation DMR-associated genes, the expected number of overlaps with the glyphosate DMR-associated genes is 78 if the DMRs are randomly distributed. Under the same assumptions, the expected number of DMR-associated genes linked to all three stress responses is 2 [(97/33602) × 712], much smaller than the observed 13 shared genes. In fact, $\chi^2$ test of independence of the three stresses is highly significant ($\chi^2 = 179.39$, *df* = 4, $p = 1.01e^{-37}$), suggesting that the

three stresses indeed induce common epigenetic changes in *A. thaliana*. Moreover, all pairwise comparison tests of independence show statistical significance (glyphosate and phosphate: $\chi^2 = 62.21$, $df = 1$, $p = 3.088e^{-15}$; glyphosate and biotic: $\chi^2 = 7.557$, $df = 1$, $p = 0.005977$; phosphate and biotic: $\chi^2 = 100.08$, $df = 1$, $p = 1.463e^{-23}$). Taken together, the comparison result shows that DMR-associated genes are not randomly distributed in the *Arabidopsis* genome and DMR-associated genes in response to one stress are not mutually independent of those involved in response to another stress.

## RESULTS AND DISCUSSION

To determine whether glyphosate injury induces changes in plant methylation, we treated *A. thaliana* with low concentration glyphosate sprays representative of real-world doses a weed on the margin of a treated field could receive. Because a plant must survive herbicide injury and produce viable seed to be a founder of an herbicide resistant population, we first identified the appropriate sub-lethal doses for glyphosate treatment of *A. thaliana*. Four-week-old *A. thaliana* rosettes were exposed to 0%, 5%, 10%, or 15% of a typical field rate of 0.9 kg acid equivalency (ae) ha$^{-1}$ glyphosate, with the 5% and 10% rates causing visible herbicide injury, but allowing for plant survival and reproduction (Fig. 1A). gDNA was collected from newly formed cauline leaves at silique maturation of four individuals from each treatment (Figs. 1B–1D). MethylC-seq of 12 total libraries (four replicates from each treatment) produced a total of 530,042,668 aligned reads resulting in genome coverages from 48× to 76× for each replicate (Table S1).

The total amount of methylated cytosines (mCs) ranged from 95 to 143 million for all replicates (Table S1) and were the most abundant in the CG sequence context (Fig. 2A), which is consistent with previous analyses of the *A. thaliana* methylome (*Cokus et al., 2008*). Only an average of 0.2% mCs was identified on the chloroplast genome, which is not methylated in *A. thaliana* (*Cokus et al., 2008*), demonstrating the fidelity of the methylC-seq protocol and data filtering. mCs were distributed across all five nuclear chromosomes. In addition, a small number of mCs were identified in the mitochondrial genome though these mCs are likely false positives because of the insertion of a region of the mitochondrial genome on chromosome 2 of *A. thaliana* eco. *Columbia* (*Lin et al., 1999*). Neither the abundance of mCs nor the relative frequency of mCs in the three different analyzed sequence contexts (CG, CHG, and CHH) significantly changed due to herbicide injury (Fig. 2A). Clustering the Pearson's correlation coefficient of the mCs did not result in treatment-specific monophyletic clades, demonstrating the relative similarity of the methylomes across all 12 individuals (Fig. S3). Taken together, these results suggest that glyphosate-induced stress did not lead to changes in overall levels of methylation in any of the three sequence contexts.

Because glyphosate did not induce global shifts in methylation levels, differences in specific location and context of DNA methylation can be potentially classified as selective responses to herbicide injury. We used Methylkit (*Akalin et al., 2012*) to compare mCs in the 5% or 10% glyphosate treatments to the control (0% treatment) to identify differentially methylated cytosines (DmCs). This approach enumerated 17,017 DmCs following 5% glyphosate treatment and 23,341 DmCs following 10% glyphosate

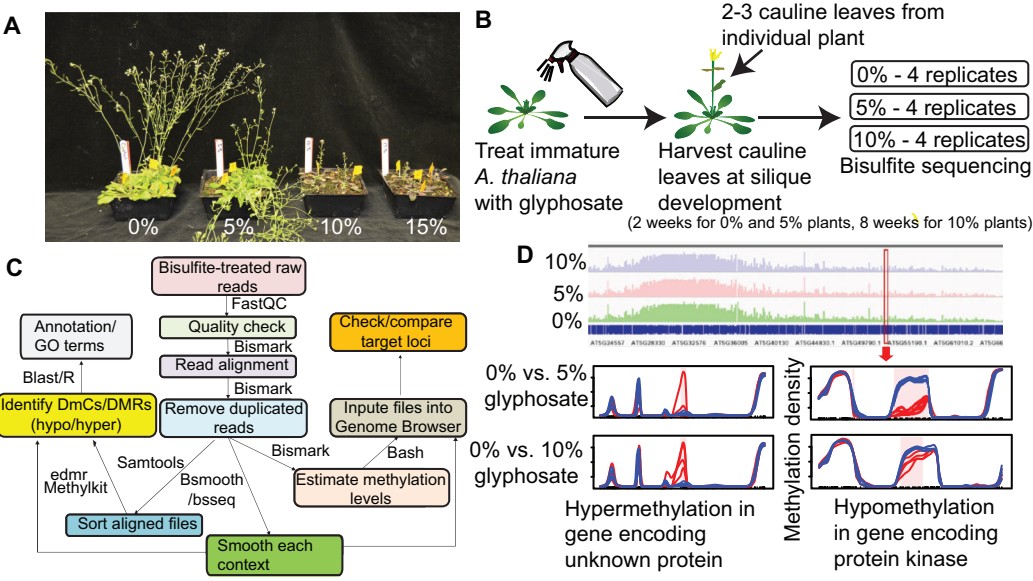

**Figure 1** **Overview of experimental pipeline.** (A) Effect of 0%, 5%, 10%, and 15% of a 0.9 kg ae ha$^{-1}$ field rate of glyphosate on *A. thaliana* floral shoot development. (B) Schematic of glyphosate treatment and tissue for harvesting gDNA. (C) Pipeline for analyzing methylC-seq data to identify differentially methylated cytosines (DmCs) and differentially methylated regions (DMRs). (D) Example of DMRs identified by bsseq. Blue lines represent methylation levels across the window of individual replicates of control plants and red lines represent methylation levels across the window of individual 5% or 10% glyphosate-treated plants.

treatment after normalizing all four replicates and applying a cutoff to only consider the top 25% highest confidence DmCs (Dataset S1). Interestingly, 10% glyphosate treatment led to substantially more DmCs than 5% glyphosate treatment in the two most abundant sequence contexts CG and CHG (Figs. 2B–2D), suggesting that the degree of herbicide injury is correlated with the context and magnitude of methylation changes. However, it is important to note that the 10% glyphosate-treated plants required an additional six weeks to reach silique maturity. Therefore, a subset of the methylation differences could be due to the increased growing time and not the herbicide treatment.

Differentially methylated regions are more strongly associated with regulatory changes in gene expression than DmCs. We called DMRs induced by both 5% and 10% glyphosate treatment using two independent approaches: a bimodal DmC distribution modeling approach using eDMR (*Li et al., 2013*), and a curve smoothing approach of DmCs using bsseq (*Hansen, Langmead & Irizarry, 2012*). All DMRs were defined as CG, CHH, or CHG based on the predominance of the sequence context of DmCs comprising the DMR. The eDMR algorithm identified 1,949 hypomethylated and 1,229 hypermethylated non-redundant DMRs following 5% and 10% glyphosate treatments (Dataset S2). More than 95% of the DMRs called by eDMR were defined by the CG sequence context (Fig. S4). Using the bsseq algorithm, 4,053 hypomethylated DMRs and 5,082 hypermethylated DMRs were identified following glyphosate herbicide injury across all three sequence contexts (Dataset S3). We excluded 70 DMRs identified in both the 5% and 10% treatment groups but conflicting in directional change of methylation (i.e., hypo in

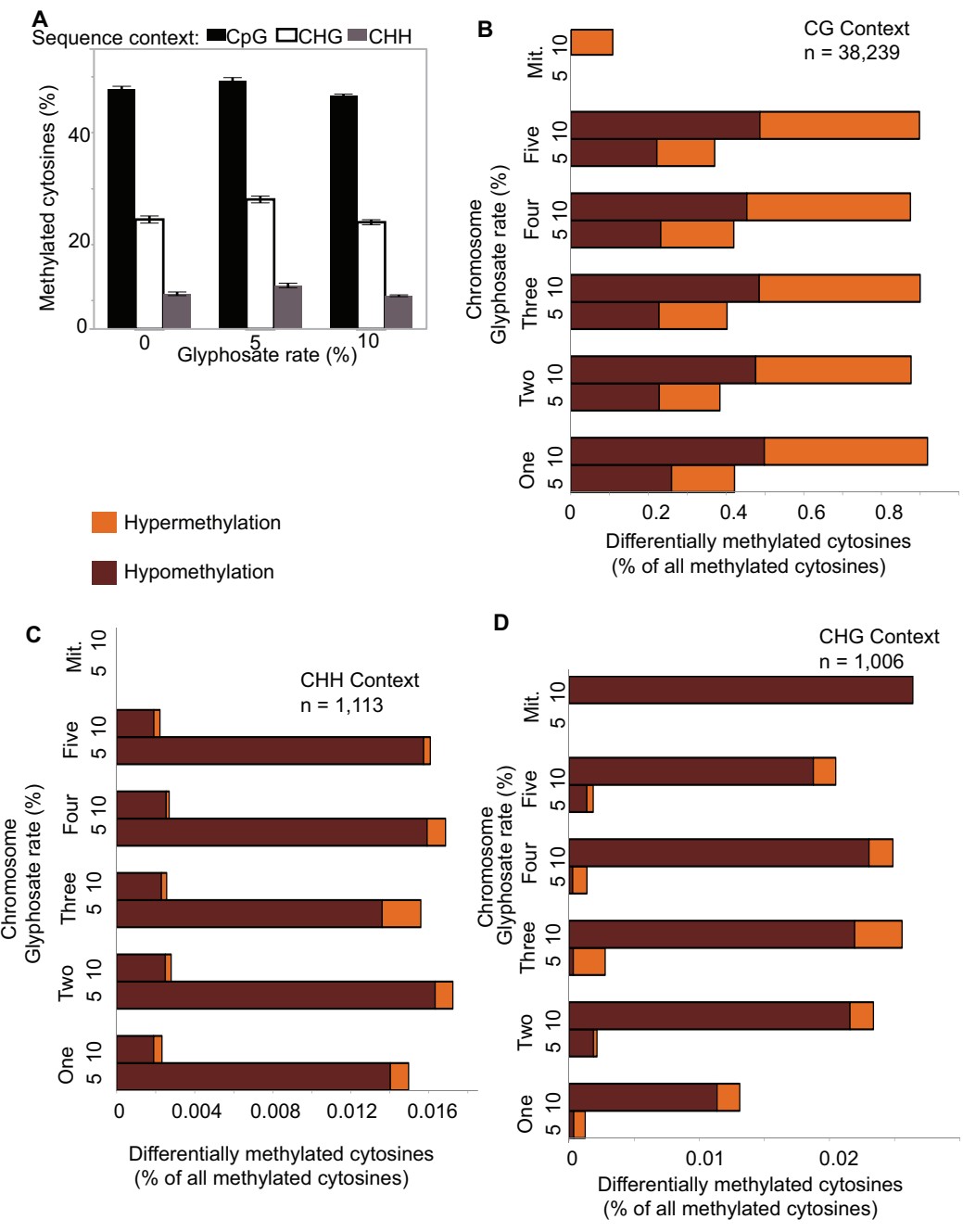

**Figure 2 Global pattern of methylated cytosines (mCs) and differentially methylated cytosines (DmCs) following methylkit pipeline.** (A) Relative abundance of mCs in the three sequence contexts (CG, CHG, and CHH) following 0, 5%, or 10% of a 0.9 kg ae ha$^{-1}$ glyphosate treatment to 4-week-old *A. thaliana* rosettes. $N = 4$ plants for each treatment. (B–D) Relative abundance of hyper- and hypo-methylated DmCs in the 5% and 10% glyphosate-treated samples compared to the 0% controls across all chromosomes in the CG (B), CHH (C), and CHG (D) sequence contexts. See Dataset S1 for list of all DmCs.

one treatment and hyper in the other treatment) and therefore unlikely to represent biologically relevant DMRs. In contrast to the eDMR data, only 78% and 51% of hypomethylated and hypermethylated bsseq DMRs, respectively, were categorized as CG

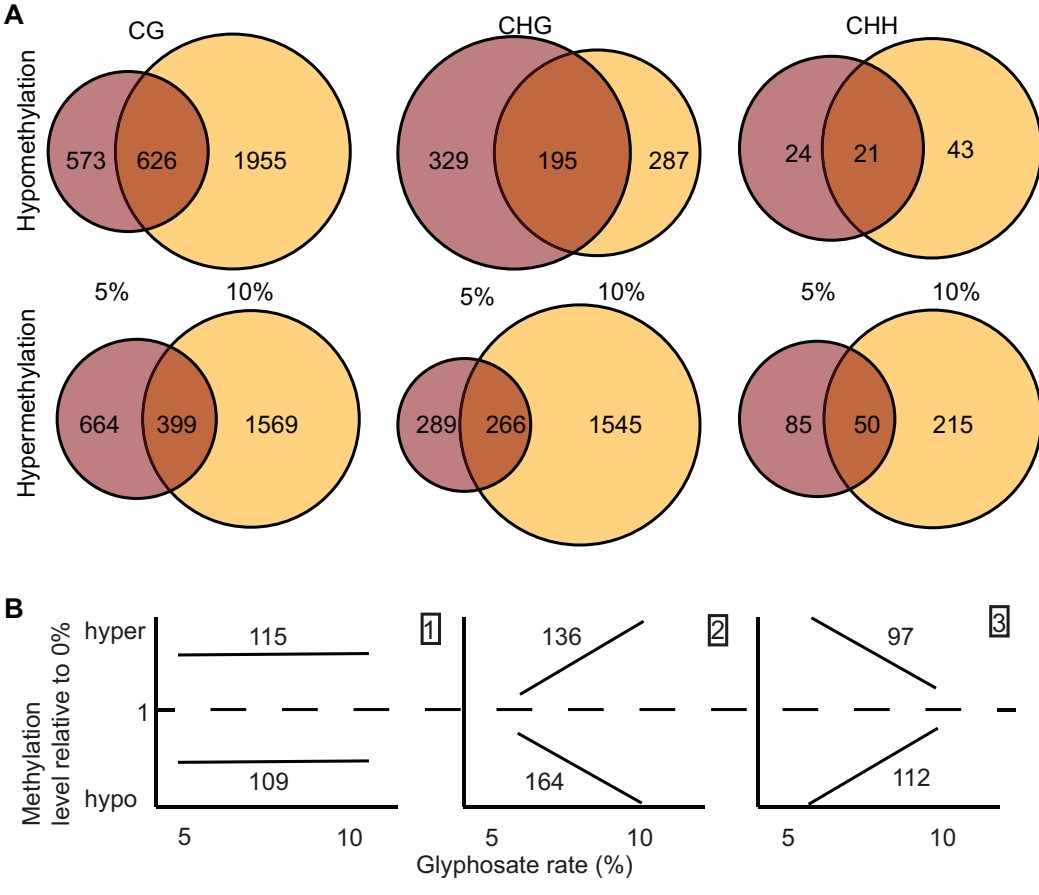

**Figure 3 Identification and dose-dependency of DMRs.** (A) overlap of DMRs between the 5% and 10% glyphosate treatment groups by sequence context. (B) Number of 5% and 10% overlapping DMRs that exhibit glyphosate dose-independent (panel 1), positive dose-dependent (panel 2), or inverse dose-dependent (panel 3) methylation responses based on non-parametric curve fitting and statistical test for significant difference between the doses at $p < 0.05$. All DMRs are described in Dataset S3. The list of dosage classification for all overlapping DMRs is available in Dataset S4. Glyphosate percentages based on a 0.9 kg ae ha$^{-1}$ rate.

(Fig. 3A). The lack of congruence between the bsseq and eDMR datasets suggests that the choice of algorithm remains a primary driver for the specificity of DMRs identified as previously suggested (*Yu & Sun, 2016*). We consider the 545 DMRs identified by both eDMR and bsseq as our highest confidence DMRs (Dataset S3). We focused on the bsseq DMRs for subsequent analyses because of the relative abundance of non-CG DMRs in this dataset and the observation that non-CG DMRs are more frequent in plants than they are in animals.

We identified 1,964 DMRs (counting both hypo- and hyper-methylated DMRs) unique to the 5% glyphosate-treated samples, and 5,614 DMRs unique to the 10% samples (Fig. 3A), which are consistent with the trend observed for DmCs in which the severity of herbicide injury correlates with the magnitude of methylome shifts. The 5,614 DMRs present only in the 10% glyphosate-treated samples were tentatively classified as positive dose-dependent DMRs (i.e., the magnitude of changes in methylation levels is positively

correlated with severity of herbicide injury) and the 1,964 DMRs unique to the 5% glyphosate-treated samples as inverse dose-dependent DMRs (i.e., more severe herbicide injury correlated with decreases in the magnitude of changes in methylation levels). The 1,557 DMRs present in both the 5% and 10% treated samples were further analyzed for dose-dependency resulting in 224, 300, and 209 DMRs classified with 95% confidence as dose-independent, positive dose-dependent, and inverse dose-dependent respectively, across both hypomethylated and hypermethylated DMRs (Fig. 3B; Dataset S4; Fig. S2; see Methods). We, therefore, conclude that for nearly two-thirds (5,914 out of 9,205) of DMRs that are positive dose-dependent, the magnitude of changes in methylation is dependent on the severity of the stress. We propose that these positive dose-dependent DMRs are the best candidates for identifying biologically relevant genomic loci that are modified in response to glyphosate stress. However, it is important to note that the 10% glyphosate-treated plants were harvested several weeks after the 5% glyphosate-treated plants to compensate for the developmental delay induced in the 10% glyphosate-treated plants (see Methods). It is possible that a number of the identified differential DMRs between the two treatment levels are due to maintenance of these DMRs being uncoupled from development and instead linked to the age of the plants.

We tentatively hypothesize for the smaller number of the inverse dose-dependent DMRs that the severity of herbicide injury following 10% glyphosate treatment prevented the plant from responding to the stress through specific alterations in DNA methylation. Interestingly, the CHH context had the most pronounced inverse dose-dependent trend, with more DmCs in the 5% treatment group than the 10% treatment group (Fig. 2), in stark contrast to the CG and CHG contexts. While some molecular mechanisms that control methylation act on all three sequence contexts [e.g., DRM 1/2 (*Cao & Jacobsen, 2002b*)], other mechanisms have distinct effects across different sequence contexts with the most pronounced differences between asymmetrical and symmetrical Cs (*Cao & Jacobsen, 2002a*; *Matzke & Mosher, 2014*). CHH sites are asymmetrical (no C in the antisense strand) in contrast to CG and CHG sites. Additionally, CHH sites were previously identified as responding uniquely to biotic stress in contrast to CG and CHG sites (*Dowen et al., 2012*). However, it is impossible to conclude that glyphosate herbicide injury alters different molecular methylation mechanisms specifically until the effects of glyphosate on methylation patterns in known methylation mutant plant lines are empirically assessed.

To elucidate which genomic regions are responding to the glyphosate stress through changes in methylation, we annotated all DMRs. A plurality of DMRs were identified in gene coding sequences and in the CG context (Fig. 4A) as expected (*Cokus et al., 2008*). The DMRs categorized as CHG or CHH were predominantly associated with transposable elements, similar to previous studies in plants (*Cokus et al., 2008*; *Li et al., 2012*). In total, 1,818 transposable elements or transposable element (TE) genes were hypermethylated and 936 were hypomethylated (Fig. 4A; Dataset S3). Transposable element-associated DMRs occurred at varying frequency across superfamily groups in response to both the 5% and 10% glyphosate treatments (Fig. S5; Dataset S5). Even though hypermethylated TEs are twice as common as hypomethylated TEs, the 936 hypomethylated TEs may

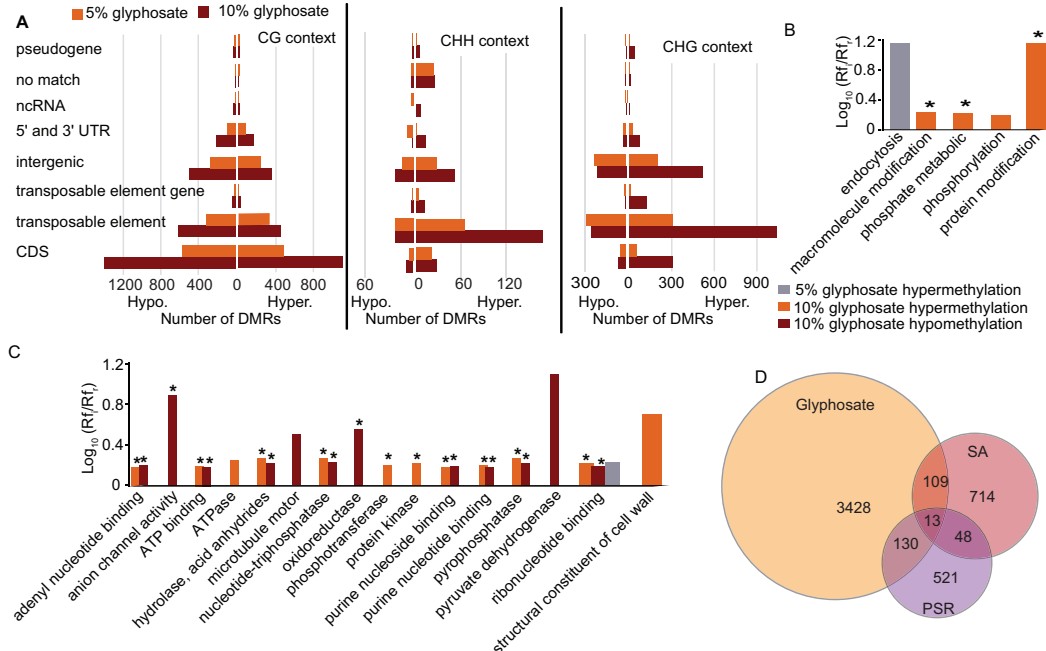

**Figure 4 Location of DMRs in the *A. thaliana* genome and comparison to DMRs induced by other stresses.** (A) Prevalence of DMRs by annotation context in the *A. thaliana* genome. See Dataset S3 for annotated list of all DMRs. (B and C) Gene ontology (GO) terms for biological process (B) and molecular function (C) enriched ($p < 0.05$) in DMR associated genes. $Rf_i/Rf_r$ represents the ratio of the relative frequency of GO terms in the input (glyphosate DMRs) to the reference (TAIR10 *Arabidopsis* genome) datasets. *indicates $p < 0.01$. Redundant GO terms excluded from the figure. See Dataset S6 for full list of GO terms with $Rf_i/Rf_r > 1.5$. (D) comparison of DMR-associated genes identified following glyphosate stress compared to DMR-associated genes previously identified as induced by biotic stress mimic (*Dowen et al., 2012*) or phosphate starvation (*Yong-Villalobos et al., 2015*). Dataset S8 lists all DMR-associated genes categorized by overlap (or lack thereof) among the three analyzed stresses.

contribute to genome destabilization due to increased transposon mobility associated with reduced transposon body methylation (*Chan, Henderson & Jacobsen, 2005*; *Mirouze & Vitte, 2014*). The number of hypomethylated TEs is less than the number of activated TEs in the DNA methylation *A. thaliana* mutant ddm1 (*Zemach et al., 2013*), but still greater than the number of TEs activated in *A. thaliana* following treatment with DNA demethylating agent azacytidine based on expression data (*Griffin, Niederhuth & Schmitz, 2016*). We, therefore, conclude that glyphosate induces hypomethylation of TEs on a scale comparable to other known treatments capable of destabilizing the *A. thaliana* genome through TE hypomethylation. The hypomethylated TEs are interesting to explore in terms of their potential for contributing to the phenomenon of gene amplification observed in several glyphosate-resistant weeds (*Gaines et al., 2010*).

An advantage to studying plant DNA methylation changes in response to glyphosate is that the biochemical mechanism of action of this herbicide has been extensively characterized. Glyphosate stops the flow of carbon through the shikimate pathway by inhibiting activity of the enzyme 5-enolpyruvylshikimate 3-phosphate (EPSP) synthase (*Amrhein et al., 1980*). Two of the seven genes in the shikimate pathway of *Arabidopsis* (*Tzin & Galili, 2010*) were differentially methylated in the CG context in response to

glyphosate stress: 3-deoxy-d-arabino-heptulosonate-7-phosphate (DAHP) synthase (At4g33510), which encodes the first enzyme (and major regulatory step for the pathway), and both forms of *shikimate kinase* (SK1, At2g21940 and SK2, At4g39540) were all hypermethylated in the 10% glyphosate-treated plants. Notably, all three of these genes were previously classified as unmethylated in *A. thaliana* (*Niederhuth et al., 2016*). EPSP synthase (At2g45300), which is also considered unmethylated in *A. thaliana* populations (*Niederhuth et al., 2016*), was not differentially methylated due to glyphosate injury (Dataset S3). Several biological processes and molecular functions were enriched in the GO terms linked to the DMR-associated genes (genes that contain DMRs in their coding sequence) (Figs. 4B and 4C; Dataset S6). The enrichment of genes associated with phosphate (including phosphorylation, phosphotransferase, protein kinase, and pyrophosphatase) was striking. This could be related to glyphosate induction of microRNAs that regulate phosphate transport pathways observed in maize (*Żywicki et al., 2015*). The glyphosate molecule is recognized by phosphate transporters (*Denis & Delrot, 1993*) and some cases of glyphosate resistance have been attributed to alterations in herbicide transport and sequestration (*Sammons & Gaines, 2014*).

As expected, the majority of the enriched GO terms were associated with genic DMRs in the CG context (Fig. S6; Dataset S6), and these included many of the terms related to phosphate. The GO terms enriched in the CHG and CHH contexts were distinct. In the CHG context, glyphosate herbicide injury was associated with alterations in methylation of cell wall-associated and hydrolase genes. Genes associated with signal reception and transduction were specifically enriched in this analysis for genic DMRs in the CHH context.

Formation and maintenance of DMRs is driven by changes in gene expression (*Secco et al., 2015*), and DMRs are sometimes associated with downstream changes in gene expression (*Li et al., 2012*) though often have no effect on gene expression (*Kawakatsu et al., 2016*). We, therefore, cannot conclude whether glyphosate is directly altering the methylation patterns of the genes associated with phosphate metabolism and the shikimate pathway or if glyphosate is altering transcription of associated genes that in turn modify the methylation states of the underlying genomic loci. Previous work showed that glyphosate caused only minor changes in immediate transcriptional activity in *A. thaliana* eco. *Columbia* (*Das et al., 2010*), but induced major effects on the transcriptome of *A. thaliana* eco. *Lansberg erecta* (*Faus et al., 2015*). Future work including paired transcriptome analyses are required to correlate methylation patterns with gene expression.

Next, we asked whether the DMRs induced by glyphosate treatment are indicative of a general stress response or potentially at least partially glyphosate specific. We compared the 3,680 non-redundant DMR-associated genes in either the 5% or 10% treatment group to DMR-associated genes that were previously identified following salicylic acid (SA) treatment, which mimics biotic stress (*Dowen et al., 2012*), or phosphate starvation stress (*Yong-Villalobos et al., 2015*). Even considering the use of different approaches and cutoffs for calling DMRs, we hypothesized that the three stresses would induce overlapping methylome alterations as a result of the plants' general response to environmental stresses

and would therefore share some DMR-associated genes. Only 13 genes were identified as being differentially methylated following all three stress treatments (Fig. 4D; Dataset S7). These 13 DMR-associated genes are top candidates for genes that are epigenetically regulated as part of a general stress response. An additional 109 and 130 glyphosate-induced DMRs overlapped with DMRs associated with biotic stress and phosphate starvation stress, respectively. In total, 3,428 (93%) of the glyphosate-induced DMR-associated genes were unique to glyphosate exposure. While the number of overlapping DMR-associated genes among the three analyzed stresses seems small, they represent significantly more than would occur by random selection from all *A. thaliana* genes (see Methods). Moreover, $\chi^2$ tests of independence of DMR-associated genes in three stresses show that stress responses are not independent of one another ($\chi^2 = 179.39$, $df = 4$, $p = 1.01e^{-37}$), and there is indeed a higher than expected number of genes involved in response to all three stresses, indicating the existence of a common methylome reprogramming pathway in *Arabidopsis* regardless of the stressors. Nevertheless, the majority of identified DMRs appear specific to glyphosate injury.

## CONCLUSION

This work identifies a large set of genes and other genomic regions epigenetically regulated in response to glyphosate herbicide injury. Determining the extent to which stresses induce patterned versus random epimutations is critical for understanding the role of DNA methylation in plant adaptation to stresses. We favor the hypothesis that the methylome changes are, at least partially, stress-specific over the hypothesis that methylome alterations are universally random because: (1) A majority of the DMRs exhibit dose-sensitive response patterns; (2) all analyzed glyphosate-induced DMRs were identified in four independent biological replicates; and (3) DMRs were enriched in gene pathways known to be affected by glyphosate exposure. Identification of transgenerationally stable DMRs and confirmation of specific DMRs directly correlated with glyphosate resistance will further clarify the role of methylome reprogramming in the evolution of herbicide resistance.

## ACKNOWLEDGEMENTS

The authors thank the Genomics Research Laboratory (GRL) at the Biocomplexity Institute of Virginia Tech for excellent MethylC-seq service.

### Funding

Seed funds were provided by the Virginia Tech College of Agriculture and Life Science and Department of Plant Pathology, Physiology and Weed Science, and additional support from the National Institute of Food and Agriculture grants nos. 2015-68004-23492 and 2013-67013-21306 (J.B.), 2015-67012-22821 (C.R.C.) and 135997 (J.H.W.). The funders had no role in study design, data collection and analysis, decision to publish, or preparation of the manuscript.

## Grant Disclosures

The following grant information was disclosed by the authors:

National Institute of Food and Agriculture: 2015-68004-23492 and 2013-67013-21306 (J.B.), 2015-67012-22821 (C.R.C.) and 135997 (J.H.W.).

## Competing Interests

The authors declare that they have no competing interests.

## Author Contributions

- Gunjune Kim performed the experiments, analyzed the data, wrote the paper, prepared figures and/or tables, and reviewed drafts of the paper.
- Christopher R. Clarke performed the experiments, analyzed the data, wrote the paper, prepared figures and/or tables, and reviewed drafts of the paper.
- Hailey Larose analyzed the data and reviewed drafts of the paper.
- Hong T. Tran analyzed the data and reviewed drafts of the paper.
- David C. Haak analyzed the data and reviewed drafts of the paper.
- Liqing Zhang analyzed the data and reviewed drafts of the paper.
- Shawn Askew conceived and designed the experiments, performed the experiments, contributed reagents/materials/analysis tools, and reviewed drafts of the paper.
- Jacob Barney conceived and designed the experiments, performed the experiments, contributed reagents/materials/analysis tools, and reviewed drafts of the paper.
- James H. Westwood conceived and designed the experiments, contributed reagents/materials/analysis tools, wrote the paper, and reviewed drafts of the paper.

## Data Availability

All sequencing data and employed scripts can be downloaded from NCBI Sequence Read Archive (SRA) BioProject ID: PRJNA322493.

## Supplemental Information

Supplemental information for this article can be found online at http://dx.doi.org/10.7717/peerj.3560#supplemental-information.

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
