# Peer review of "Herbicide injury induces DNA methylome alterations in Arabidopsis"

_PeerJ, doi:10.7717/peerj.3560_

## Round 0.1 · original submission · Major Revisions

Your manuscript entitled “Herbicide injury induces dose-dependent DNA methylome reprogramming in Arabidopsis” has been seen by two qualified reviewers. Based on their recommendation and my own detailed assessment, I feel this manuscript is potentially suitable for publication in PeerJ, but there are some issues that need to be addressed as well as optional analyses that would strengthen the manuscript. Both reviewers felt the manuscript was technically sound, but each reviewer had concerns about the interpretation of the results that should be addressed by the authors. Below I have summarized the major issues raised by the reviewers, though the authors should provide responses to all of the queries.

1. The supplemental figures appear to be missing
2. The suggested analyses by reviewer 1 are not essential, but would greatly improve the biological interpretations of the manuscript and can be done relatively quickly without major changes to the manuscript.
2. Reviewer 2 felt the differences in collection times for the 5% and 10% glyphosate treatments complicates the interpretations of dose dependent affects. The authors should address this issue and justify the rational for choosing this sampling scheme. If the authors can demonstrate that this is not an issue, the interpretations of a dose dependent methylation effect can be included. If not, they should be removed as suggested by reviewer 2.

·

Basic reporting

Overall, the paper reporting of the paper is adequate. There is one major issue, as it appears the supplemental figures are missing form the submission. There are some minor comments on statements and also additional information about the sequencing that are needed.

Major issue:

1) Figures S1-S5 appear to be missing and are needed for a full evaluation.

Minor issues:

2) Line 56: “The primary mechanism of epigenetic regulation of gene expression is through cytosine methylation.”

This is an unsupportable statement. In Arabidopsis, a significant proportion of genes are unmethylated and many thousands are regulated instead by histone marks like H3K27me3 instead (see Zhang et al 2007). At the same time, CHG and CHH methylation are also strongly associated with the presence of histone marks like H3K9me2 (Du et al, 2012).

3) Line 58: “CG methylation predominantly in gene rich regions”

CG methylation in Arabidopsis is found predominantly in gene-poor regions alongside CHG and CHH methylation (see Lister et al 2008, Cokus et al 2008). This is also true across plant species.

4) Non-conversion rates should be reported for each sample.

Experimental design

The overall experimental design is good. More than sufficient replication and sequencing for each condition is provided. Sufficient information is provided to reproduce the results.

Validity of the findings

The bioinformatic analysis conducted is typical of the field and sufficient for publication. Two aspects of the analysis and their reporting are problematic and must be corrected corrected for publication. I also have one minor comment on the interpretation of their results.

Major

1) The paper mentions methylation of the mitochondria and shows this in Figure 2. These are almost certainly false positives and should be removed. The Columbia accession used in this study has a recent and nearly complete insertion of the mitochondrial genome on chromosome 2 with 99% identity to the mitochondrial genome (Lin et al 1999). This is the likely source of the mitochondria methylation the authors observe. WGBS of Arabidopsis accessions that do not have this insertion, such as Landsberg erecta, do not show DNA methylation of the mitochondria as observed in the Columbia accession.

2) The authors use the Das et al 2009 dataset and look for overlap of DMRs with 101 herbicide responsive genes, finding overlap for 49 genes. However, Das et al 2009 found only 16 DEGs in response to glyphosate treatment. The list of 101 genes were identified on their ability to distinguish between different types of herbicide treatments and include genes that are differentially expressed under other classes of herbicide treatment, but not glyphosate. Claims regarding this dataset should be modified to correctly correspond to the actual Das et al 2009 data. The authors might also consider looking at other datasets if available, such as Faus et al 2015 (BMC Plant Biology) which also looked at differential expression in response to glyphosate treatment in the Landsberg erecta accession and found a larger number of differentially expressed genes.

Minor

3) Line 277-279. “The relative abundance of hypermethylated transposable elements suggests that glyphosate herbicide injury does not lead to global Arabidopsis genome destabilization due to increased transposon mobility associated with reduced transposon body methylation”

I do not fully agree with the authors that the predominance of hypermethylated DMRs within transposons suggests global instability due to transposition is unlikely. A significant portion (936) were hypomethylated and as the authors suggest, potentially active. In the absence of additional data or analysis, it is not clear what the effects would be. While this number is much less than the number of TE genes that are transcriptionally reactivated in Arabidopsis DNA methylation mutants like ddm1 (Zemach et al 2013 Cell), it is also greater than the number of TEs that are reactivated after treatment with DNA demethylating agents (Griffin et al 2016 G3). This claim would require at the minimum transcriptional data from treated and untreated samples.

Additional comments

It becoming increasingly apparent that stresses can induce epigenomic changes in plants. While these changes typically revert in following generations (Wibowo et al 2016), some may persist in following generations (Jiang et al 2014, Wibowo et al 2016). Clarke et al examine the methylomes of plants exposed to the herbicide glyphosate and so are the first to extensively study the genome-wide epigenomic effects this deliberate man-made stress. I find that some of the results, such as the rather specific enrichment of GO terms in phosphorous related pathways to be intriguing. I would suggest to the authors that the biological relevance of the results could be improved with perhaps deeper analyses.

Suggested analyses and discussion

1) The DMR analysis is potentially very interesting and should be explored in greater depth. Specifically, I am interested in the previous methylation status of genes containing DMRs. In plants, genes can be categorized by their methylation profile (Takuno & Gaut 2012, Takuno & Gaut 2013, Niederhuth et al 2016). Genes methylated only in the CG context are referred to as gene-body methylated and are typically associated with constitutively active genes. Genes methylated in CHG and CHH sites are typically suppressed, while unmethylated genes show a range of expression profiles and include those genes that are more environmentally responsive. This is optional, but a clearer interpretation of the methylation changes could be deduced by examining the types of DMRs (CG, CHG, CHH) and what classes of genes these are occurring in. As an example, I did a quick comparison of CG DMRs from Dataset 2 to genes classified by their methylation status in Niederhuth et al (2016). 1316 of 2992 genic CG DMRs were located to gene-body methylated genes versus 90 in CHG/CHH methylated genes. This indicates that much of the genic CG differential methylation is probably occurring in genes already methylated in that context. By considering the DMR context and the previous methylation status of the gene, a more biologically relevant interpretation of these methylation differences can be made. For example, hypermethylated CHG/CHH DMRs in a gene-body methylated or unmethylated gene would be indicative of potential suppression of that gene’s expression. In contrast, hypermethylated CG DMRS located in a gene-body methylated gene which is already methylated in the CG context is likely less biologically relevant or indicative of effects on expression. Similarly, hypermethylated CHG/CHH DMRs in genes that already contain this methylation is unlikely to change the expression of that gene, as these are already suppressed, whereas hypomethylated CHG/CHH DMRs in a gene previously methylated in these same genes might be associated with activation of that gene.

2) The enrichment of phosphorous related GO term in DMRs is potentially very interesting and would benefit from a deeper exploration. It is possible that this enrichment is simply due to a high proportion of those genes already having methylation and thus more likely to gain or lose additional methylation. Across Arabidopsis accessions DMRs are known to be enriched in certain GO terms (Schmitz et al 2013, Kawakatsu et al 2016) and genes classified as gene-body methylated or non-CG methylated are also enriched in various GO terms (Takuno & Gaut 2013, Niederhuth et al 2016). I know that several of the reported enriched phosphorous related terms are also enriched amongst gene-body methylated genes, but so are many other GO terms related to metabolism. I am surprised and interested then by how specific the enrichment for phosphorous related terms is. At the very least, I would like to see the authors examine the GO term enrichment broken down by the category of DMR (CG, CHG, CHH), as it may be that each of these categories is enriched in different types of GO terms (Schmitz et al, 2013). Some discussion into the previously methylated state of the gene would also be helpful. For example, are the phosphorous related hyper-methylated DMRs occurring in genes that are already methylated (such as gene-body methylated genes) or is a gain of methylation observed at previously unmethylated genes, which would indicate de novo methylation.

3) For the Shikimate pathway genes that are differentially methylated, I would like to see it mentioned in the text the context of this methylation and its direction (hyper/hypo).


References (this review)

Cokus, S. J., S. Feng, X. Zhang, Z. Chen, B. Merriman, C. D. Haudenschild, S. Pradhan, S. F. Nelson, M. Pellegrini and S. E. Jacobsen (2008). "Shotgun bisulphite sequencing of the Arabidopsis genome reveals DNA methylation patterning." Nature 452(7184): 215-219.

Das, M., J. R. Reichman, G. Haberer, G. Welzl, F. F. Aceituno, M. T. Mader, L. S. Watrud, T. G. Pfleeger, R. A. Gutierrez, A. R. Schaffner and D. M. Olszyk (2010). "A composite transcriptional signature differentiates responses towards closely related herbicides in Arabidopsis thaliana and Brassica napus." Plant Mol Biol 72(4-5): 545-556.

Du, J., X. Zhong, Y. V. Bernatavichute, H. Stroud, S. Feng, E. Caro, A. A. Vashisht, J. Terragni, H. G. Chin, A. Tu, J. Hetzel, J. A. Wohlschlegel, S. Pradhan, D. J. Patel and S. E. Jacobsen (2012). "Dual binding of chromomethylase domains to H3K9me2-containing nucleosomes directs DNA methylation in plants." Cell 151(1): 167-180.

Faus, I., A. Zabalza, J. Santiago, S. G. Nebauer, M. Royuela, R. Serrano and J. Gadea (2015). "Protein kinase GCN2 mediates responses to glyphosate in Arabidopsis." BMC Plant Biol 15: 14.

Griffin, P. T., C. E. Niederhuth and R. J. Schmitz (2016). "A Comparative Analysis of 5-Azacytidine- and Zebularine-Induced DNA Demethylation." G3 (Bethesda) 6(9): 2773-2780.

Jiang, C., A. Mithani, E. J. Belfield, R. Mott, L. D. Hurst and N. P. Harberd (2014). "Environmentally responsive genome-wide accumulation of de novo Arabidopsis thaliana mutations and epimutations." Genome Res 24(11): 1821-1829.

Kawakatsu, T., S.-shan C. Huang, F. Jupe, E. Sasaki, Robert J. Schmitz, Mark A. Urich, R. Castanon, Joseph R. Nery, C. Barragan, Y. He, H. Chen, M. Dubin, C.-R. Lee, C. Wang, F. Bemm, C. Becker, R. O’Neil, Ronan C. O’Malley, Danjuma X. Quarless, N. J. Schork, D. Weigel, M. Nordborg and J. R. Ecker (2016). "Epigenomic Diversity in a Global Collection of Arabidopsis thaliana Accessions." Cell 166(2): 492-505.

Lister, R., R. C. O'Malley, J. Tonti-Filippini, B. D. Gregory, C. C. Berry, A. H. Millar and J. R. Ecker (2008). "Highly integrated single-base resolution maps of the epigenome in Arabidopsis." Cell 133(3): 523-536.

Niederhuth, C. E., A. J. Bewick, L. Ji, M. S. Alabady, K. D. Kim, Q. Li, N. A. Rohr, A. Rambani, J. M. Burke, J. A. Udall, C. Egesi, J. Schmutz, J. Grimwood, S. A. Jackson, N. M. Springer and R. J. Schmitz (2016). "Widespread natural variation of DNA methylation within angiosperms." Genome Biol 17(1): 194.

Schmitz, R. J., M. D. Schultz, M. A. Urich, J. R. Nery, M. Pelizzola, O. Libiger, A. Alix, R. B. McCosh, H. Chen, N. J. Schork and J. R. Ecker (2013). "Patterns of population epigenomic diversity." Nature 495(7440): 193-198.

Takuno, S. and B. S. Gaut (2012). "Body-methylated genes in Arabidopsis thaliana are functionally important and evolve slowly." Mol Biol Evol 29(1): 219-227.

Takuno, S. and B. S. Gaut (2013). "Gene body methylation is conserved between plant orthologs and is of evolutionary consequence." Proc Natl Acad Sci U S A 110(5): 1797-1802.

Wibowo, A., C. Becker, G. Marconi, J. Durr, J. Price, J. Hagmann, R. Papareddy, H. Putra, J. Kageyama, J. Becker, D. Weigel and J. Gutierrez-Marcos (2016). "Hyperosmotic stress memory in Arabidopsis is mediated by distinct epigenetically labile sites in the genome and is restricted in the male germline by DNA glycosylase activity." Elife 5.

Zemach, A., M. Y. Kim, P. H. Hsieh, D. Coleman-Derr, L. Eshed-Williams, K. Thao, S. L. Harmer and D. Zilberman (2013). "The Arabidopsis nucleosome remodeler DDM1 allows DNA methyltransferases to access H1-containing heterochromatin." Cell 153(1): 193-205.

Zhang, X., O. Clarenz, S. Cokus, Y. V. Bernatavichute, M. Pellegrini, J. Goodrich and S. E. Jacobsen (2007). "Whole-genome analysis of histone H3 lysine 27 trimethylation in Arabidopsis." PLoS Biol 5(5): e129.

·

Basic reporting

The article is written well and easy to follow.
Raw data are publicly available and relevant processed data sets are supplemented.
Minor points:
1. In the Introduction section, lines 56-57, the authors write that “The primary mechanism of epigenetic regulation of gene expression is through cytosine methylation.” I would rephrase the wording to “A prominent mechanism of…”, as strong DNA methylation Arabidopsis mutants did not alter gene expression stronger than other epigenetic mutations, like H3K27me3.
2. The statement in line 58 “…CG methylation predominantly in gene rich regions…”, is not correct. In many plant genomes, definitely in ones that do not methylate genes at all, CG methylation is predominantly targeted to TEs. Hence, it is more accurate to say that in plants non-CG methylation, i.e. CHG and CHH, is predominantly targeted to transposons and repeats, where CG methylation can also be frequently found in genic sequences.
3. The resolution of Figure 1A needs to be enhanced.
4. Line 382, publication year should change from 2009 to 2010

Experimental design

In this study the goal of Clark et al. was to investigate changes in DNA methylation in Arabidopsis plants treated with glyphosate. Mapping Arabidopsis methylomes post herbicide treatment have been sporadically done before. One study also mapped methylation in wheat treated with glyphosate, though only at limited genomic loci (Narderin et al., 2015; should be cited by the authors). Like the authors clearly mentioned in their paper, glyphosate is a predominant herbicide currently used in agriculture, with 35 plant species already developed resistance to it. Hence, what is the global as well as the specific effect of glyphosate has on DNA methylation, and what roles these alterations may play in plant response and adaption are valid and interesting questions to ask.

Clark et al., methylome data is pretty solid and reliable. They mapped methylomes of four biological replicates for each sample, control and glyphosate treated plants. The sequencing data has high coverage (48x-76x per replicate) and high bisulfite (C to U) conversion rate (98.8%). Using these datasets, the authors could reliably identify differentially methylated cytosines and regions between control and glyphosate treated plants.

In an effort to substantiate the association between glyphosate and methylation the authors tested for a dosage methylation effect to two glyphosate concentrations (5% and 10%). This is a good strategy, however according the harvest time for 10% was significantly longer (6 weeks difference) than the one of the control and 5% treated plants. This period of time was chosen by the authors to allow harvesting treated plants at equivalents developmental stages (10% glyphosate treated plants are developmentally delayed). DNA methylation in Arabidopsis has been reported before to change a bit during development, however the mechanism for it is unknown 1. Hence, by harvesting 10% treated plants 6 WEEKS AFTER 0% and 5% plants have been harvest, could allow more time for methylation changes to further establish themselves in the genome. Therefore, to my opinion the authors cannot consider the different glyphosate percentage treatments as different dosage conditions. Additionally, the author chose to harvest plants several weeks (2-8) after glyphosate treatment, time that allow treated plants to change their morphology and developmental progress. This harvesting strategy makes it hard to differentiate between cause and effect, i.e. does methylation plays a primary role in plant response to glyphosate or does it the consequence of the responding of plants to glyphosate, a point that is of great significance to the objective of this study. A better way to study the role of glyphosate treatment in plants would have been harvesting plants and mapping their methylation at two time points, soon after glyphosate treatment (24-48hr) and at a later stage. In that way, we could learn whether methylation is being regulated early in the treatment and persist or inherited during plant development or does it appear in later stages which would imply for a secondary effect. To summarize this issue, I would say that treating plants with two glyphosate concentrations that each was harvest at different time points, basically cancel both advantages (concentration and time) and does not help to understand neither dosage effect nor dynamic regulation of DNA methylation.

Finally, in the paper the authors nicely associate the differential methylation regions in treated plants to various genomic annotations. A further association of methylation differences to published methylation profiles of Arabidopsis plants mutated in positive and negative methylation pathways, such as RNA directed DNA methylation (e.g. drm2), H3K9me2 (e.g. cmt3) and DNA demethylation (e.g. ROS1), could provide a mechanistic understanding of the methylation dynamic processes happening in glyphosate treated plants.

To summarize, most of the findings of this study are focused on whether methylation is changing, and if so where in the genome, in response to glyphosate treatment. This is sufficient for publication in PeerJ, however any suggestions or hints by the authors of an active and specific mechanism for DNA methylation in response to glyphosate treatment need to be removed. For example, I would change the word “reprogramming” in the title to “alterations”, and remove the whole sentence in lines 25-28 and 31 in the abstract that is based on ‘dose-dependent’ methylation effect and implies that plants “CAN” modulate methylation based on the severity of stress.

Validity of the findings

1. Clark et al., produced a good amount of well executed methylome data. They process and analyze it using adequate aligning and statistical methods. Therefore, I believe that their findings of differential methylation in glyphosate treated plants is real.

2. Based on my comment in the Experimental Design section, I would request to remove all references in the text that suggest dose-dependent methylation effect, including from the title, abstract and conclusion sections.

3. Lines 293-295 “The enrichment of (differentially methylated) genes associated with phosphate (metabolism)… was striking.” Genic methylation was previously shown to be directed by gene expression, hence the changes in methylation in phosphate related genes could be the result of changes in their expression level and not necessarily the cause of their expression alterations. The authors actually mention this argument themselves in the next paragraph (lines 300-302). To test the relation with transcription the authors correlate their methylation data with transcriptional published data acquired by a different group. This correlation found 49 out of the 101 alternatively expressed genes to be also differentially methylated, which is a bigger overlap than expected by random chance (if you include all methylated and expressed genes, >20K genes), which might suggest of a link between both gene lists. However, “there was no clear association of hypo or hypermethylation with upregulation or downregulation of gene expression selectively”, which could suggest that methylation changes in glypohsate injured plants is no directed by transcription. This might be true, however it is worth mentioning that the glyphosate treatments in Clarke et al., (methylation) and Das et al., (RNA) were done quite differently. Clarke et al., sprayed glyphosate on mature (four weeks old) plants and harvest cauline leaves from injured plants 2-8 weeks post treatment, whereas Das et al., sprayed 14-day old seedling plants and harvest rosette “leaves without visible symptoms” 24hr post treatment. Accordingly, it is reasonable to assume that these two studies will result with different gene regulation outcomes and therefore lay a significant doubt for its relevancy. Like I mentioned in the Experimental Design section, to test for cause and effect between methylation changes and transcription alterations, the authors should have profiled plants early after glyphosate treatment (like done by Das et al.,) and later in development.
4. The conclusion of the authors that Arabidopsis methylome is reprogrammed in a stress specific way (lines 330-340) is based on incorrect interpretation of their findings: 1) their methylation data cannot be interpreted as a dose-dependent response; 2) biological replicates are being used to identify real DMRs and not necessarily to determine if they are specific or random. If any, the fact that methylation is changed in both directions (up and down), both in genes and transposons, and in thousands of genes, would imply that the glyphosate has a quite general effect on methylation; 3) the enrichment of DMRs in gene pathways known to be affected by glyphosate exposure, is an interesting observation, but like I mentioned above (point # 3) methylation changes in those genes could be transient and the consequence of their transcriptional alteration.
5. In lines 321-323 the authors concluded that the number of overlapped DMR-associated genes among glyphosphate and two other stresses is significantly higher than would occur by random selection from all A. thaliana genes. The authors based their calculation and conclusion on the assumption that DMRs in the three stresses are randomly located in all Arabidopsis genes (Methods lines 175-178). Instead of assuming random distribution of DMRs the authors should have checked this assumption in their data, as it is actually reasonably to assume that methylation would fluctuate where it is normally found, i.e. within methylated genes. This is a critical point because that if stress induced DMRs are indeed located specifically within methylated genes (<8000 genes; Takuno and Gaut, 2012), then the number of differentially methylated genes overlapped between glyphosphate and either of the two other stresses or with both of them would shift from being overrepresented to underrepresented (~25-60% less than expected by random chance), a result that would cancel the authors’ conclusion in lines 326-327, of an “existence of a common methylome reprogramming pathway in Arabidopsis regardless of the stressors.”
6. Lines 227-230 talks about dosage dependent methylation effect in CG and CHG. Interestingly, CHH DMCs showed an opposite dose dependent trend, where 5% treated plants had much more DMCs than 10% treated ones. Molecular mechanisms for targeting methylation (RdDM or H3K9me2) and removal of methylation (DME or ROS1) are usually function similarly on all or multiple methylation contexts, hence to see a separate trends of methylation reprograming for different methylation contexts is an unusual result that needs to be mention and discuss. This discrepancy could be explained by the various harvesting time points of 5% and 10% treatments, which supports the reason for why we cannot identify the different treatments solely by their glyphosate concentration.

Additional comments

Your highly replicative deep methylome data clearly shows that DNA methylation changes due to glyphosate treatment. Your analyses nicely show the level and genomic distribution of these changes for each of the methylation contexts. I am not convinced that the changes are dose dependent or selective. Nevertheless, even by removing or correcting these conclusions from/in the text, I believe that your paper is warrant to be published and your methylation data would be a good resource for the scientific community.

I marked your paper for major revisions, not because I request additional experiments but because of the necessary of removal several main unsupported interpretations written in your paper.

---

## Round 0.2 · Major Revisions

Your revised manuscript has been assessed by two reviewers. Most of the previous comments were adequately addressed, but reviewer 1 has raised an additional concern about analysis of DMRs. I feel this work is suitable for publication in PeerJ after the authors address this issue as well as a few minor criticisms.

·

Basic reporting

1. I find the reporting of methylated data in Table S1 a bit confusing. It seems that the authors are reporting the number of methylated cytosines from each read and then a percent methylation, however it is not clear how the percent methylation was calculated and the reporting of the number of methylated cytosines in the reads, without the inclusion of total cytosines (unmethylated + methylated) is unconventional. Is the percent methylation reported here a weighted methylation, i.e. mC/(C+mC) or is this the percentage of CG sites that were called as methylated after say application of a binomial test (see Schultz et al, 2012 Trends in Genetics for discussion of metrics for calculating methylation levels)?

2. In Materials and Methods, the section on bsseq, there is discussion of "less stringent" and "more stringent" parameters. These parameters should be specified for reproducibility.

3. In Materials and Methods, "custom Perl scripts" are mentioned. Please make all scripts publicly available, preferably through a repository like Github, Bitbucket, etc for reproducibility.

4. In Materials and Methods, section Gene Ontology Enrichment Analysis, please discuss how association of DMRs with features was performed (i.e. overlap with TEs/Genes, overlap with upstream 1Kb, 2Kb, etc).

5. In Results and Discussion, line 259-260. The claim that "non-CG DMRs are more relevant for methylation changes in plants than they are for animals", should be removed or replaced. Non-CG methylation exists and appears to have important role in specific animal tissues/cell types. Furthermore, its debatable whether or not non-CG is more relevant than CG methylation in plants. Arabidopsis mutants that eliminate most of non-CG methylation in Arabidopsis typically fare better than Arabidopsis met1 mutants, which eliminates most of CG-methylation.

6. In Results and Discussion, line 267-268, the authors state "more severe herbicide injury correlated with decreases in methylation levels" in refrence to the inverse DMRs. Are all 1964 of these DMRs hypomethylated or is this a mix of both hyper and hypo? If it is a mix, then the wording of this statement needs to be changed to reflect that.

7. In Results and Discussion, line 307-312. The authors discuss their results in comparison to Zemach et al 2013 and Griffin et al 2016 as a comparison of hypomethylation. Neither one of those studies called DMRs or calculated the number of DMRs in a way that would allow the sort of comparison as worded in the text. They did look at the number of differentially expressed TEs. In my previous review, I put these number of hypomethylated DMRs into the context of reactivated TE genes to make the point that about TE expression and methylation. This section should be reworded to make it clear that this comparison of hypomethylated genes is in comparison to expressed TEs after loss of DNA methylation.

Experimental design

1. In Materials and Methods, lines 179-181, the authors compare the number of DMR-associated genes following glyphosate treatment to other stresses. I am concerned that this analysis is not valid based on the large differences in how the various papers call DMRs and map them to genes. For example, Secco et al 2015 (phosphate-starvation dataset) called CNN DMRs (all contexts) as opposed to calling them individually for CG, CHG, and CHH (this paper). This would result in the collapsing of overlapping CG, CHG, and CHH DMRs into a single DMR (which is appropriate), this may or may not affect the number of overlapped genes. They also filtered their DMRs by number of differentially methylated sites, which would also reduce the number of DMRs called. Looking at the size of the author's DMRs, while most are large enough to include multiple sites, a few are as small as 3bps and thus likely based on a single DmC. Differences in how DMRs are associated with genes (strict overlap, within 2Kb, etc) would also greatly impact the number of associated genes.

Validity of the findings

1. I have concerns regarding the large difference in results between eDMR and bsseq in the DMR calling and how this also integrates with the calling and reporting of DmCs. The authors reported that >95% of DMRs called by eDMR. They then call DMRs using bsseq, resulting in a greater proportion of non-CG DMRs. However, when one looks at the DmCs called by Methylkit, ~95% of these are in the CG context, so it is not surprising then that eDMR called 95%+ CG DMRs. Furthermore the overlap of DMRs between the two methods is rather low (~17% of eDMR and ~6% of bsseq) which suggests to me that at the very best, one of these methods is performing poorly. I am also not convinced by the author's rationale for focusing on the bsseq data because of the greater abundance of non-CG DMRs.

Additional comments

I thank the author's for their response. After careful rereading and the inclusion of the supplemental figures I have additional concerns regarding the DMR calling and the comparison of DMRs between stresses and that are drawn from this. I also have additional minor comments regarding the reporting of the materials and methods.

·

Basic reporting

I would like to thank the authors that generally corrected or commented effectively to most of my first remarks.

I still would like to ask the authors to better emphasize to the readers about their experimental design, in which they harvested 10% treated plants 6 weeks after 0% and 5% plants. They should mention it in the text as soon as they talk about their experiment and not to leave this detail to the end of the DMR paragraph in the ‘results and discussion’ section. They should also add this detail to Figure 1B that illustrates their experimental system. It is likely that the additive effect in methylation is real, but not having control plants for the growth period of six weeks between 0% and 10% plants, is just against the standard requirement in experimental biology. From their methods section it sounds like that the 4 biological replicates were grown simultaneously, which means that any alterations in growing conditions (not necessarily evidenced by the authors) during the extended 6 weeks of growth, would influenced only 10% treated plants, and all of its biological replicates.

Minor issues

1. Line 90, ‘vernalized’ should change to stratified
2. Figure 1D. ‘ecoding’ should change to ‘encoding’

Experimental design

see Basic reporting.

Validity of the findings

See Basic reporting.

---

## Round 0.3 · accepted · Accept

The reviewers have re-evaluated your revised manuscript. The revisions have addressed the previous concerns and I feel it is now suitable for publication in PeerJ.

·

Basic reporting

The reporting of the results is clear and suitable for publication.

Experimental design

The authors have justified their rationale for the DMR comparisons with other stress conditions and addressed my concerns.

Validity of the findings

The authors have noted the difference in DMR calling between bsseq and eDMR in the paper and addressed my concerns.

Additional comments

I thank the authors for their efforts in addressing my comments.